# Microbiologically Influenced Corrosion of a Pipeline in a Petrochemical Plant

**Mahdi Kiani Khouzani** [1] **, Abbas Bahrami** [1]**, Afrouzossadat Hosseini-Abari** [2]**,**
**Meysam Khandouzi** [1] **and Peyman Taheri** [3,*]

[1] Department of Materials Engineering, Isfahan University of Technology, Isfahan 84156-83111, Iran;
    mahdi.kiani1996@gmail.com (M.K.K.); a.n.bahrami@cc.iut.ac.ir (A.B.); shokraneh_84@yahoo.com (M.K.)

[2] Department of Biology, Faculty of Sciences, University of Isfahan, Isfahan 817463441, Iran;
    afrouz_hosseini1985@yahoo.com

[3] Delft University of Technology, Department of Materials Science and Engineering, Mekelweg 2,
    2628 CD Delft, The Netherlands

* Correspondence: p.taheri@tudelft.nl; Tel.: +31-15-278-2275

**Abstract:** This paper investigates a severe microbiologically influenced failure in the elbows of a buried amine pipeline in a petrochemical plant. Pipelines can experience different corrosion mechanisms, including microbiologically influenced corrosion (MIC). MIC, a form of biodeterioration initiated by microorganisms, can have a devastating impact on the reliability and lifetime of buried installations. This paper provides a systematic investigation of a severe MIC-related failure in a buried amine pipeline and includes a detailed microstructural analysis, corrosion products/biofilm analyses, and monitoring of the presence of causative microorganisms. Conclusions were drawn based on experimental data, obtained from visual observations, optical/electron microscopy, and Energy-dispersive X-ray spectroscopy (EDS)/X-Ray Diffraction (XRD) analyses. Additionally, monitoring the presence of causative microorganisms, especially sulfate-reducing bacteria which play the main role in corrosion, was performed. The results confirmed that the failure, in this case, is attributable to sulfate-reducing bacteria (SRB), which is a long-known key group of microorganisms when it comes to microbial corrosion.

**Keywords:** microbiologically influenced corrosion (MIC); sulfate-reducing bacteria (SRB); failure; amine pipeline

## 1. Introduction

Pipelines and fittings are considered to be strategic components in petrochemical plants, especially when it comes to sustainable and safe operation. Amongst different fittings, elbows, mostly with angles of 90° or 45°, are the most used ones. Elbows can experience severe damage in the case of dramatic changes in the flow pattern. As far as materials are concerned, carbon steels are widely used for the transmission of petroleum products and water. On the grounds that carbon steel is not remarkable as a corrosion-resistant alloy, external corrosion of buried carbon steel pipelines/fittings is always considered a major issue in petrochemical plants. The consequences of failures in buried pipeline/fittings are severe in most cases, owing to the fact that repair and replacement of damaged pipes are costly and difficult, in the sense that the damaged area needs first to be reached by digging into the ground. This implies more time, needed for maintenance operations, associated with longer downtimes for an installation. Other than that, failure of pipes/fittings can cause environmental contaminations, imposing health risks to the public [1–4]. Pipeline corrosion is influenced by a large number of factors such as soil resistivity, soil chemistry, temperature, pH, aeration, and microorganisms.

For buried underground carbon steel pipelines several measures are normally taken to minimize the corrosion risk, including applying coating and putting the cathodic protection system into service. Any damage in the coating or improper selection of cathodic protection settings results in corrosion progression over long-term service [5,6].

Amongst the different possible corrosion mechanisms, microbiologically induced corrosion (MIC), also named microbiologically influenced corrosion, is the one closely related to the activity of living microorganisms in the soil, including microalgae, bacteria, archaea, and fungi. The economic costs associated with microbiologically influenced corrosion in buried pipelines in oil, gas, petrochemical industries, power plants, and other chemical plants are enormous. Microbiologically influenced corrosion can take place in environments and working conditions where there is no other corrosion taking place, or it can take place in combination with other corrosion failures. More importantly, microorganisms can accelerate the kinetics of anodic/cathodic corrosion reactions, in such a way that they can be viewed as "catalytic" entities. Microorganisms are more known to induce a localized attack, including dealloying, pitting, localized galvanic corrosion, and stress corrosion cracking [1,7–9]. MIC-induced problems in buried pipelines/equipment have attracted lots of attention all around the world [1,9]. MIC, in any form, starts with the formation of a biofilm on the metal substrate [1], within which planktonic cells get attached to the substrate of steel, where they grow, reproduce, consume nutrients and produce an extracellular polymer substance (EPS) in addition to other metabolic products that generate a localized corrosion cell and collectively build the biofilm. It is also well known that microorganisms can use chemicals as nutrient sources and oxidize them [10–12]. Microbial corrosion of steel is divided into aerobic and anaerobic corrosions, with the latter known to induce a higher corrosion rate, while the former is associated with a lower corrosion rate [13]. Anaerobic microorganisms can also grow and control corrosion, in the absence of oxygen. When it comes to biocorrosion of steels, iron-reducing bacteria (IRB), iron- and manganese-oxidizing bacteria, acid-producing bacteria, and sulfate-reducing bacteria (SRB) are microorganisms which are recognized to have more detrimental effects. The latter is known to be one of the main culprits that cause severe biodegradation of the external surface of buried pipelines. As far as SRB-induced biocorrosion of the interior of pipes is concerned, the low content of oxygen, together with stagnant liquid inside the pipelines, provide desirable conditions for SRB to grow. In addition, SRB can appear under deposits of soil, water, hydrocarbons, chemicals, etc. The kinetics of corrosion in steels under the influence of microorganisms, including SRB, can be up to ten times faster than the kinetics in the absence of microorganisms [6]. It is reported that SRB could get electrons directly from Fe when there is no availability of organic carbon sources. This could occur when the soil acts as a barrier between the steel surface and the external environment [6]. Figure 1 shows a schematic drawing of the SRB function in MIC. SRB can use elemental iron as an energy source or electron donor (anode reaction). Furthermore, in order to produce energy and maintain electroneutrality, SRB uses sulfate as a terminal electron acceptor in the reaction at the cathode (following reaction 1 as microorganism) [2,3,13]. Corrosion products which are formed by reactions are given here, in Reactions 2, 3 and 4:

$$SO_4{}^{2-} + 9H^+ + 8e^- \rightarrow HS^- + 4H_2O \tag{1}$$

$$Fe^{2+} + HS^- \rightarrow FeS + H^+ \tag{2}$$

$$Fe^{2+} + 2OH^- \rightarrow Fe(OH)_2 \tag{3}$$

$$Fe(OH)_2 \rightarrow Fe_3O_4 + H_2 + 2H_2O. \tag{4}$$

This research investigates a severe MIC-induced failure in a buried amine plant installation. The amine treatment process is, in fact, a "sweetening" process in which excess carbon dioxide and hydrogen sulfide are removed from sour or natural gas. Even though there are lots of studies on microbial corrosion, there are few field investigations on buried pipelines available. This investigation studies MIC in a real industrial case. Although there are many microorganisms in the environment

since the major failure mechanism is the presence of SRB, the purpose of this paper is to prove the role of SRB. The findings of this study could be very useful when it comes to identifying and preventing MIC in buried pipelines.

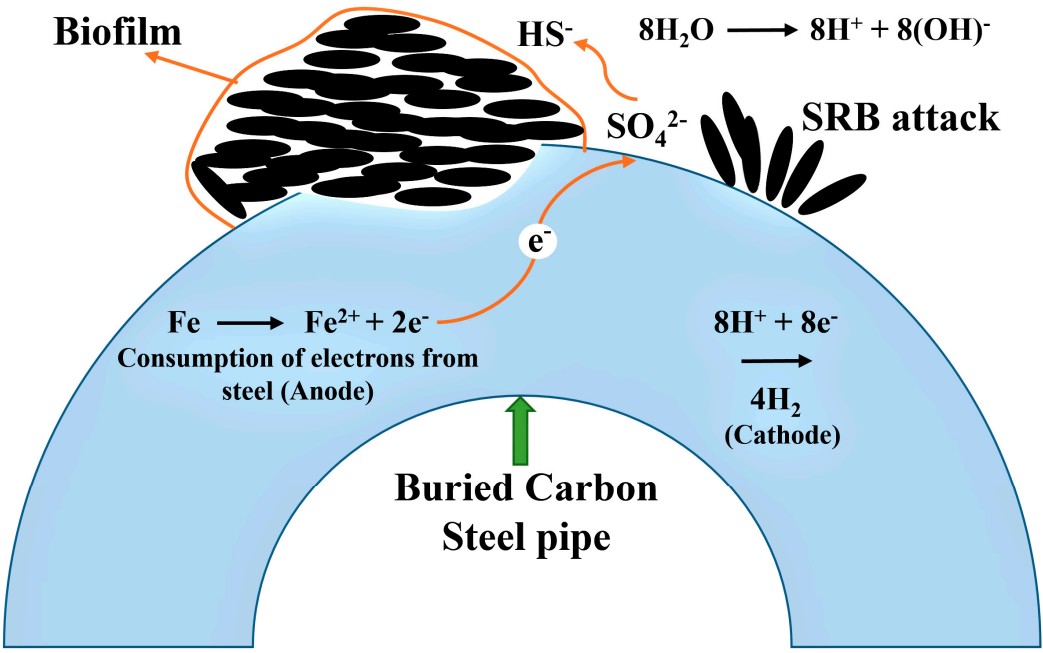

**Figure 1.** A schematic view of the function of sulfate-reducing bacteria (SRB) in microbiologically induced corrosion (MIC).

## 2. Materials and Methods

### 2.1. Case Background

This work studies the severe failure of carbon steel elbows in a buried amine pipeline of a petrochemical plant. Massive failure was observed after less than only five years of operation. The plant underwent a major overhaul to replace damaged elbows. According to the design specifications, the elbow was made of SA 234-WPB carbon steel and was placed underground. Table 1 shows the chemical–physical analysis and grain size of the soil. Results show that the soil contained 9.0% coarse (>76.2 mm) particles. This is an important factor when it comes to the water retention capability of the soil. In addition to that, the particle size distribution affects oxygen diffusion and solute transport, with both having significant implications for the corrosion rate. The pH of the soil was around 8.5. Microbial diversity is noticeably affected by pH, with acidic soil harboring the least diverse communities and neutral soil harboring the most diverse ones. pH values below 5.5 are generally a sign of lower contents of calcium, magnesium, and phosphorus. The sulfate/sulfide content of the soil is also important because sulfur stimulates the metabolism of SRB in the soil. The soil contained some chloride ions as well. Chloride makes the environment more aggressive since it damages the passive oxide film on the surface and causes localized corrosion [2,14].

**Table 1.** Specifications of the onsite soil (taken from the laboratory of the plant according to ASTM D4318-17e1 and ASTM D4972-18).

| Parameters | Percentage of Coarse Particles (>76.2 mm) | pH (%) | SO$_3$ (%) | Cl (%) | Salt Content (%) | Plasticity Index | Liquid Limit (%) |
|---|---|---|---|---|---|---|---|
| Sample result | 9.0 | 8.5 | 0.14 | 0.03 | 0.05 | N.A. | 17 |
| Specifications limit | - | - | - | - | - | ≤6% | ≤25% |

## 2.2. Experiment

The failure analysis, in this case, is mostly based on a microstructural investigation, using routine metallographic techniques, including optical microscopy (OM, Nikon, Tokyo, Japan) and scanning electron microscopy (SEM), coupled with energy dispersive X-ray spectroscopy (EDS, Philips, Eindhoven, The Netherlands). Failed elbows were studied in this investigation. Optical emission spectroscopy (OES, Quantometer 34000-ICP-OES System, Thermo Scientific, Arun, UK), was used to determine the elemental composition of failed samples. Macroimages were taken using a Stereo Microscope Nikon SMZ 800 (Nikon, Tokyo, Japan). For metallography, samples were prepared by grinding the surface with 180 to 1200 grinding papers, followed by polishing with diamond paste (3 and 1 μm). Etching was performed with Nital 60% (the solution contained 1–10 mL nitric acid with 100 mL ethanol). Phase identification was conducted using an X-ray diffractometer (XRD, Philips, Eindhoven, The Netherlands), using Cu K-alpha radiation. To study the presence and effects of SRB, some surface cuts of the corroded tube and the soil from the damaged area were inoculated in an SRB-specific medium (the modified form of API RP-38 medium), which contained Na lactate 3.5 g, peptone meat extract 1.2 g, $MgSO_4$ 0.9 g, $NaSO_4$ 0.3 g, $K_2HPO_4$ 0.3 g, $CaCl_2$ 0.06 g, $Fe(NH_4)_2(SO_4)_2$-$6H_2O$ 0.39 g, and ascorbic acid 0.1 g in 1000 mL distilled water [15]. In order to prevent $O_2$ penetration, liquid paraffin was added after inoculation and incubated at 25 °C for 20 days in anaerobic conditions. Soil samples were collected from the vicinity of the damaged area. Samples were collected from ten different spots with an approximate weight of 10 g each. Collected soil samples were stored in plastic bottles. In order to assess the microbial activity, the redox potential of the as-received soil before inoculation in an SRB-specific medium and that after inoculation was measured using a multi-parameter analyzer (Consort C 535 model, Belgium).

## 3. Result and Discussion

### 3.1. Visual Observations

Figure 2 shows a typical failed elbow, taken out of the service. Both the inner and outer surfaces were covered with red/yellow–brown and, in some spots, black rust layers. The elbows were heavily damaged at the outer surface. Large pits and perforations were seen on the surface. Pits on the exterior of the failed elbows were at different stages, with some at the early stages, while a few have turned into big holes during service. While the outside part of the elbows had undergone severe localized corrosion, the inside part hardly showed any pitting, see Figure 3, indicating that the pitting had started from the outer surface. There were hardly any signs of surface irregularities (e.g., erosion or surface deformation) other than severe surface pitting.

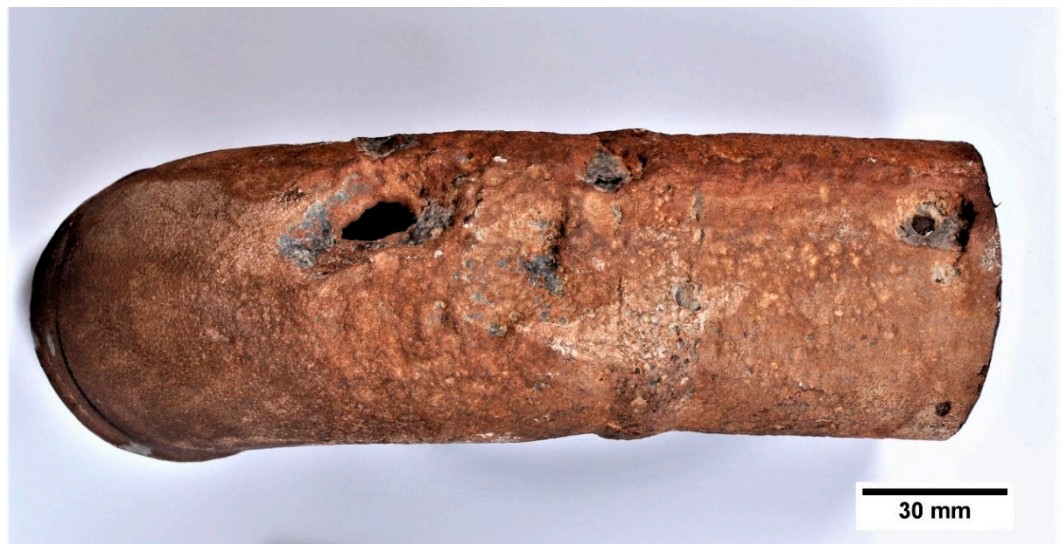

**Figure 2.** An example of the failed elbows.

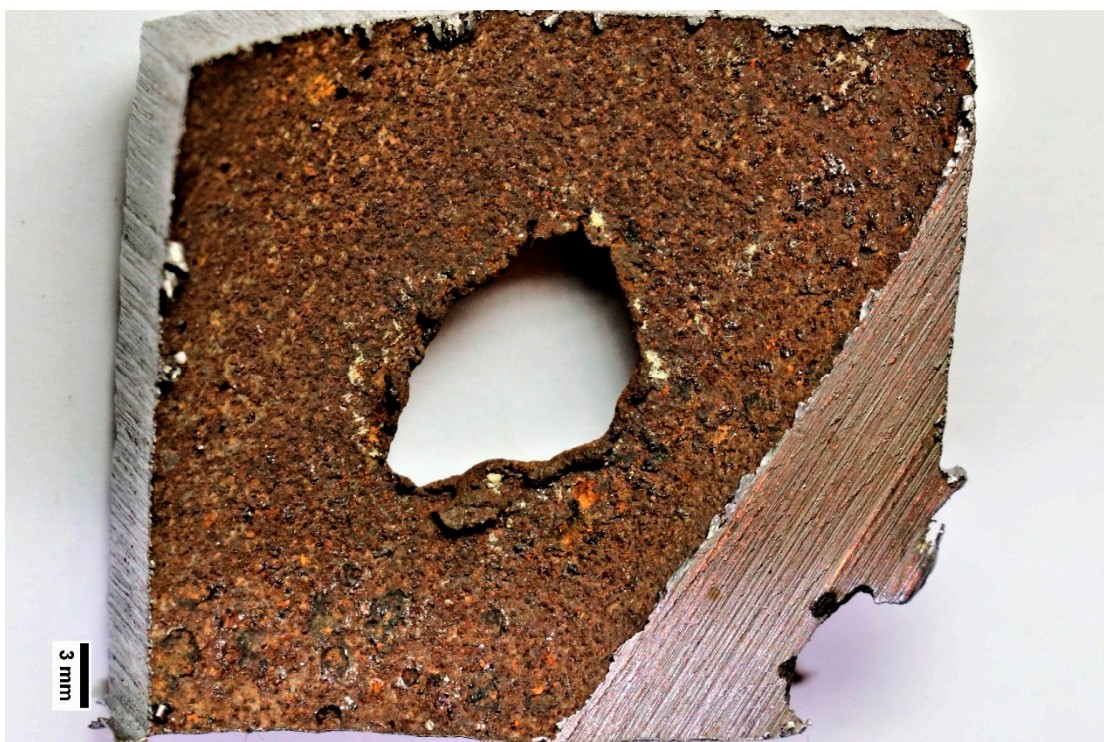

**Figure 3.** An inner surface of the failed elbows.

### 3.2. Chemical Analysis

OES, or Quantometer, was used to measure the chemical compositions of the alloy. Results were compared with standard ASTM A234 WPB to check the conformity of the chemical composition of the alloy with the standard values. Table 2 shows the chemical composition of the sample and its corresponding ASTM standard values. The results show that the chemical composition of the failed elbow is almost in accordance with standard values.

**Table 2.** The chemical composition of the sample and its corresponding ASTM standard values.

| Elements | C | Mn | P | S | Si | Cr | Mo | Ni | Cu | V | Nb |
|---|---|---|---|---|---|---|---|---|---|---|---|
| Sample | 0.20 | 0.64 | 0.02 | 0.01 | 0.23 | 0.03 | ≤0.05 | 0.01 | 0.01 | - | - |
| ASTM-A234 WPB | 0.3 | 0.29–1.06 | 0.05 | 0.058 | 0.1 min | 0.4 | 0.15 | 0.4 | 0.4 | 0.08 | 0.02 |

### 3.3. Morphology of Pits

Figure 4a shows the microstructure of the failed elbow at the inner surface. The observed elongated ferritic–pearlitic microstructure is an indication that the alloy was in the as-rolled state. As mentioned before, there was no indication of pitting at the inner surface. Figure 4b shows the microstructure of a cross-section at the outer surface. Severe perforation and deep pits were observed at the outer surface. Furthermore, pits were covered with corrosion products, while in some cases, corrosion products had a layered structure.

### 3.4. Analysis of Corrosion Products/Biofilm at the Outer Surface

Figure 5 shows the typical features found inside pits at the outer surface. This figure clearly shows that the surface was covered with a biofilm. Biofilms, consisting of complex communities/colonies of microorganisms and EPS, can dissolve oxygen and create gradients of pH, resulting in localized forms of corrosion, such as crevice corrosion and pitting [10]. Three noticeable features were seen in the biofilm:

(i)   Rod-shaped products, with approximate length of 50 μm;
(ii)  Colonies of spheres, with their size ranging from 10 to 100 μm;
(iii) Flower-like surface features, all shown in Figure 5.

Rod-like features were, statistically, more frequently seen than the other two morphologies. The distributions of all the mentioned morphologies were rather homogeneous, i.e., all features were seen in each and every pit.

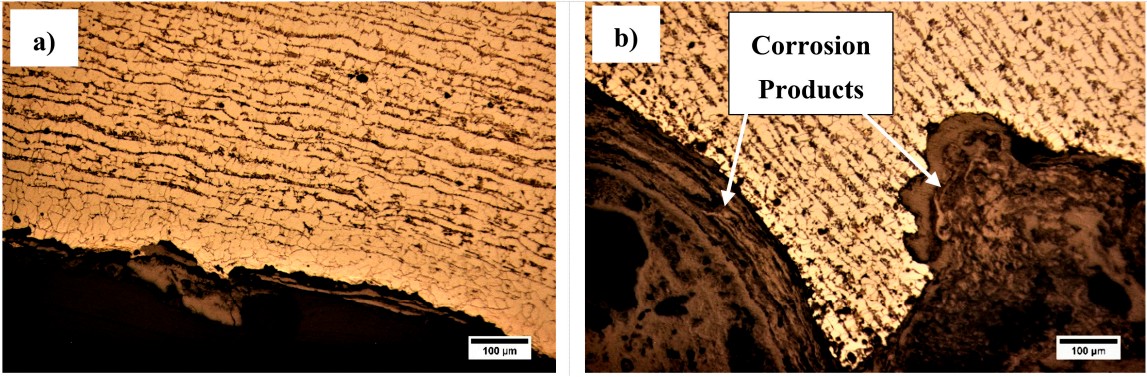

**Figure 4.** The optical microscopy (OM) microstructures of the (**a**) inner surface and (**b**) the outer surface.

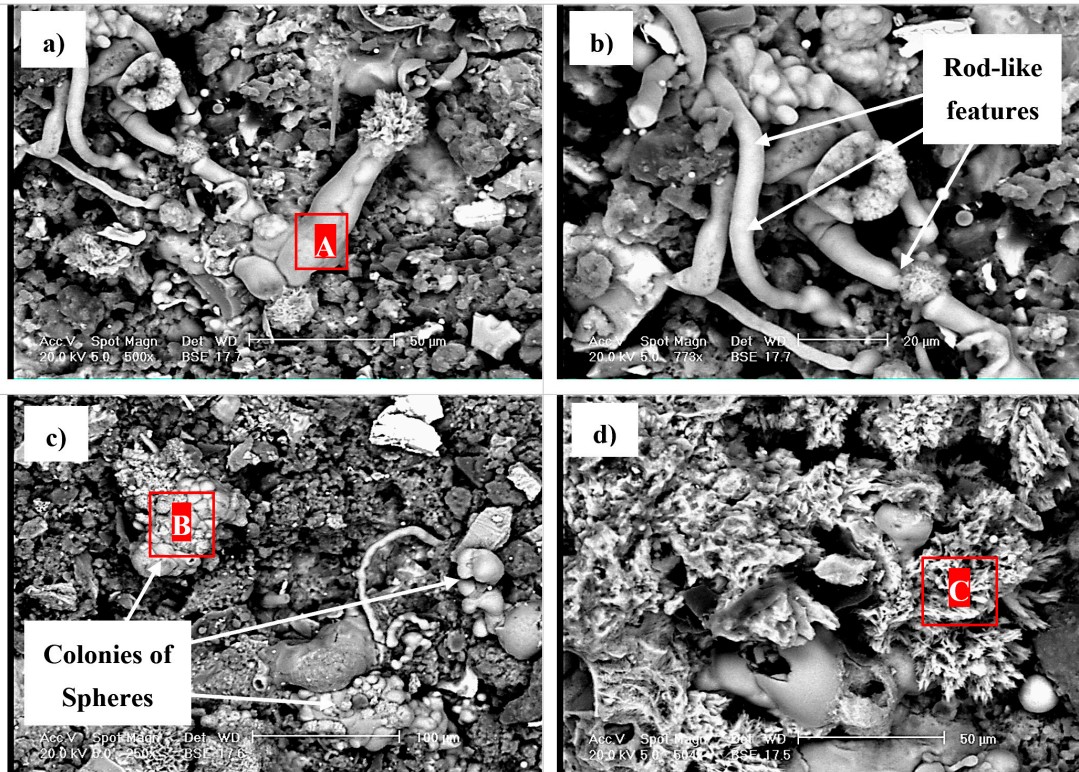

**Figure 5.** Scanning Electron Microscope (SEM) photomicrographs showing biofilms (or surface deposit) and features inside biofilms, i.e., (**a**) and (**b**) rod-like features, (**c**) colonies of spheres, and (**d**) flower-like features.

Figure 6 shows the results of EDS analyses of rod-like, spherical, and flowerlike corrosion products. In terms of the chemistry, there was no meaningful difference between these features, inferring that the difference between these features is only morphological. The presence of sulfur in all the EDS spectra is an indication that the observed features are SRB, which is an anaerobic bacterium. SRB is known to be the primary factor of MIC. Organic compounds attracted to the surface of steel make it an ideal environment for the colonization of microorganisms [2]. Amongst different microorganisms, SRB is known to have excellent adaptability to extreme conditions. Biochemical energy for the growth of SRB is obtained by reducing sulfate ($SO_4^{-2}$) to sulfide ($S^{-2}$) in the presence of natural organic compounds. At the same time, with the absorption of atomic hydrogen (H) on the surface of iron, it will gradually disappear [16]. Overall, the major detrimental metabolic activities of SRB are mostly related to its ability to:

(i)　use natural organic compounds as electron donors,
(ii)　oxidize hydrogen,
(iii)　utilize aromatic and aliphatic hydrocarbons, and
(iv)　reduce sulfate to sulfide [17].

Some sulfate-reducers can oxidize methane in anaerobic reactions. Another unique characteristic of SRB is its ability to survive when exposed to oxygen, even though its growth is inhibited in such a case. It is worth mentioning that the spherical geometry of particles might be an indication of the presence of iron (II) carbonate ($FeCO_3$) surrounded by a mixture of iron oxides and iron sulfide (FeS). From an engineering point of view, the importance of microorganisms' nature is that they have quite a large variation in size. They are also homogeneously distributed, potentially very fast-growing, and they have significantly grown with perceivable restraints in nutrient availability induced by an overpopulating of microorganisms [11,17].

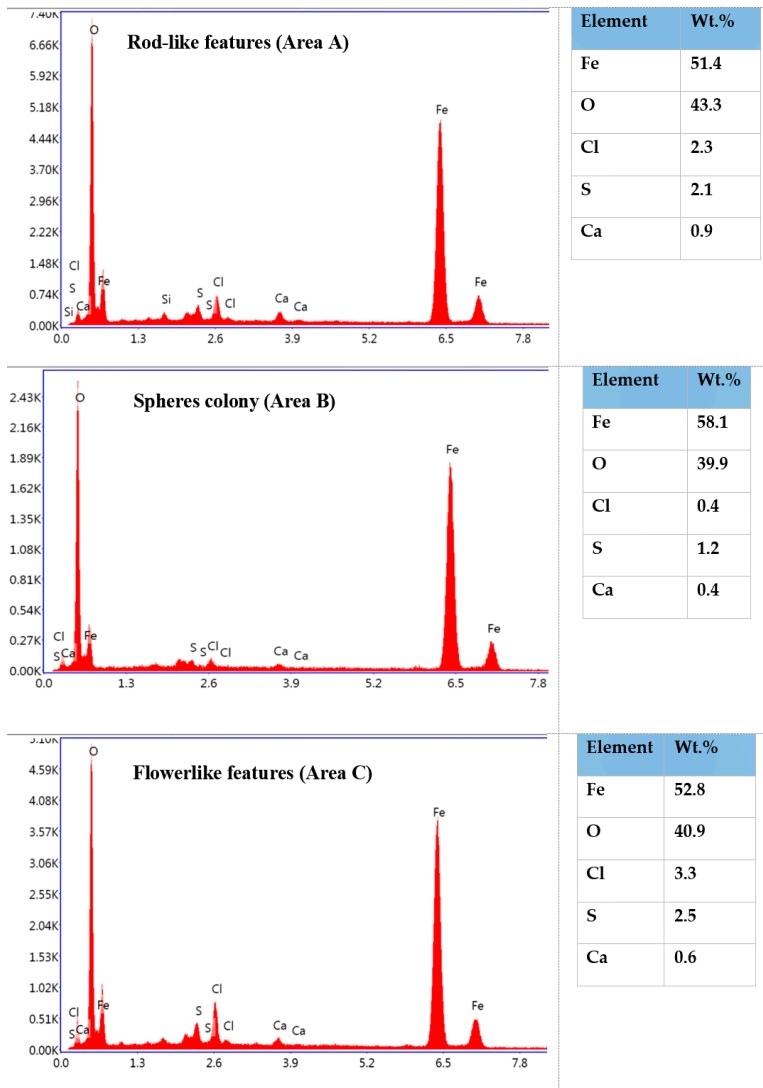

**Figure 6.** Energy dispersive X-ray spectroscopy (EDS) analyses of areas shown in Figure 5, related to rod-like, spherical, and flower-like features (*x*-axis: Energy (KeV), *y*-axis: a.u.).

Figure 7 represents a closer look at the biofilm in the outer surface in which microorganisms are distributed. Biofilm is a self-produced firmly-attached EPS surface layer. In fact, the stability of microorganisms is dependent on the biofilm, given that the biofilm provides structural support for microorganisms and, to a great extent, protects them from environmental, physical, and chemical stresses. Biofilms could host a variety of different microorganisms including fungi, bacteria, eukaryotes, and archaea, with each having a different metabolic ability [2]. The SRB, in particular, has a vigorous metabolism, producing a certain amount of EPS (in fact, often there are also other microorganisms present that are mainly responsible for EPS production), which rapidly adhere to the surface of the metal and build a biofilm [14,18]. The biofilm, in this case, has a porous nature, making the ion/nutrition exchange much easier. EPS are composed of charged polymers and can significantly contribute to the accumulation of different chemical species inside EPS. In the case where organic acids are produced by microbial species, mineral dissolution rates are expected to increase [19,20]. The results of EDS analysis show that elements oxygen, sulfur, calcium, chloride, and iron were present in the biofilm. Figure 8 shows typical XRD patterns of corrosion products, taken from the surface of the pipe. The results confirmed the formation of a significant amount of oxide-hydroxide (FeOOH), iron oxide, and iron sulfide. The latter is a typical corrosion product when SRB is the controlling microorganism.

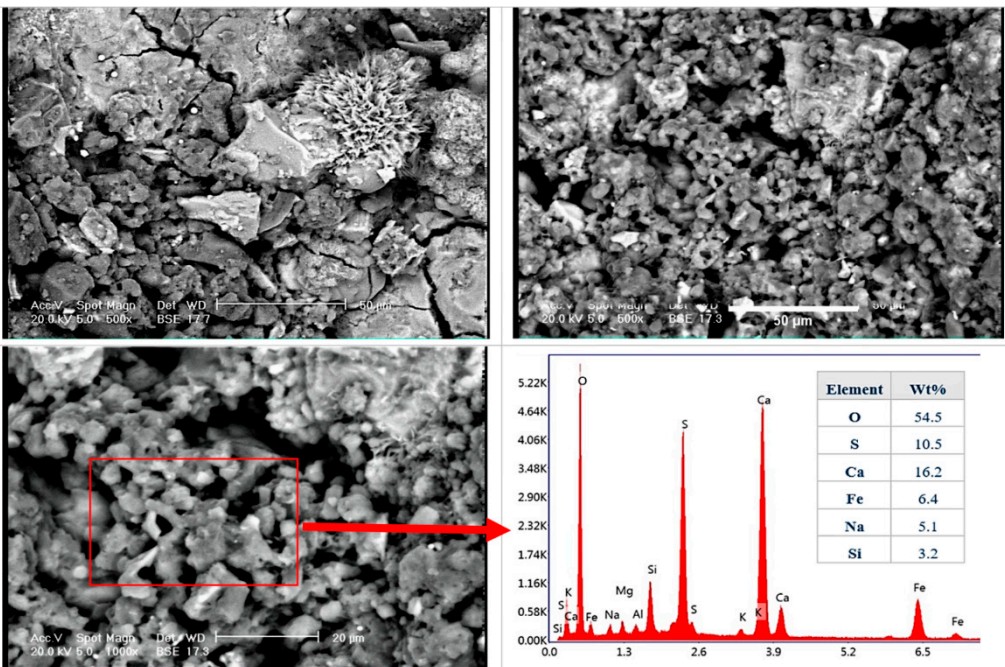

**Figure 7.** Examples of SEM images of biofilms together with EDS analysis.

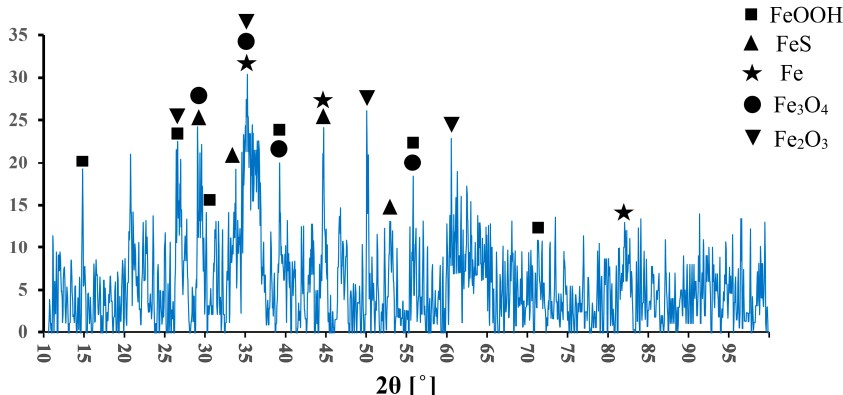

**Figure 8.** An X-ray diffraction analysis of the surface of the pipe.

### 3.5. Analysis of Corrosion Products at the Inner Surface

Figure 9 shows SEM images of corrosion products at the inner surface. As mentioned earlier, there was hardly any sign of localized corrosion. The inner surface in some areas was covered with a porous scale-like layer intermingled with some globular $\alpha$-FeOOH particles, see Figure 9a,b. Other observed morphologies of corrosion products were plate-like micaceous crystalline features, see Figure 9c, and honeycomb-like structures, see Figure 9d, with both being typical morphologies of $\gamma$-FeOOH. Therefore, overall, the inner surface was covered with different iron oxide-hydroxides and possibly other oxide products. Figure 10 shows the EDS analysis of the areas shown in Figure 9. There was no sign of sulfur and chloride, unlike what was seen at the outer surface. It appears that the failure has started from the outer pipe surface, and corrosion products at the inner surface were caused by the liquid inside the pipeline.

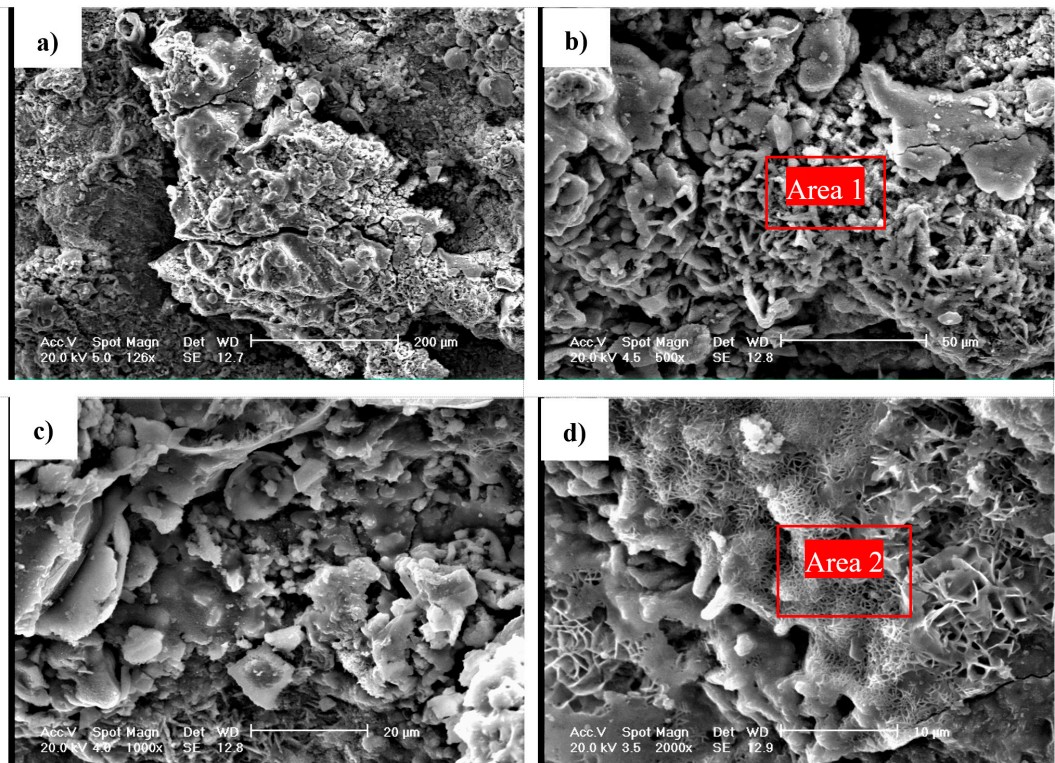

**Figure 9.** Different morphologies of the corrosion products at the inner surface: (**a**) and (**b**) scale-like layer intermingled with some globular $\alpha$-FeOOH particles, (**c**) plate-like micaceous crystalline features, and (**d**) honeycomb-like structures.

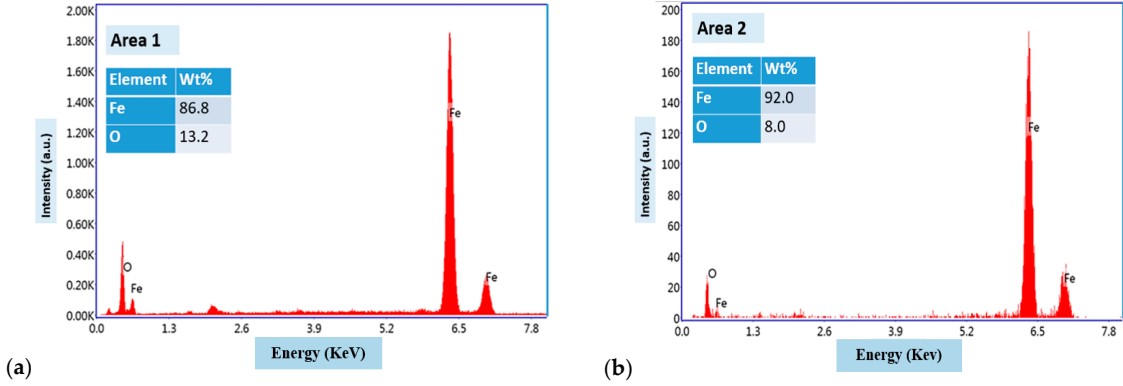

**Figure 10.** EDS analyses of (**a**) area 1 and (**b**) area 2, shown in Figure 9.

*3.6. SRB Monitoring at the Corroded Site*

Surface scratches from the corroded elbows and samples of soil from the corrosion area were inoculated in SRB-specific media and incubated in anaerobic conditions for 20 days. As shown in Figure 11, the black precipitation is created due to SRB growth and $H_2S$ and FeS production in the media. It is noticeable that the SRB-induced black precipitation was only created on the corroded surface of the failed elbow, and there was no sign of black precipitation on the cut surface, see Figure 11b. The formation of SRB cells was confirmed by SEM images, see Figure 12, and EDS spectra, see Figure 13. SRB is resistant to environmental stresses such as pH and temperature variations, chloride concentration, and high values of pressure. They form biofilms and create massive corrosion products by deteriorative mechanisms [21,22].

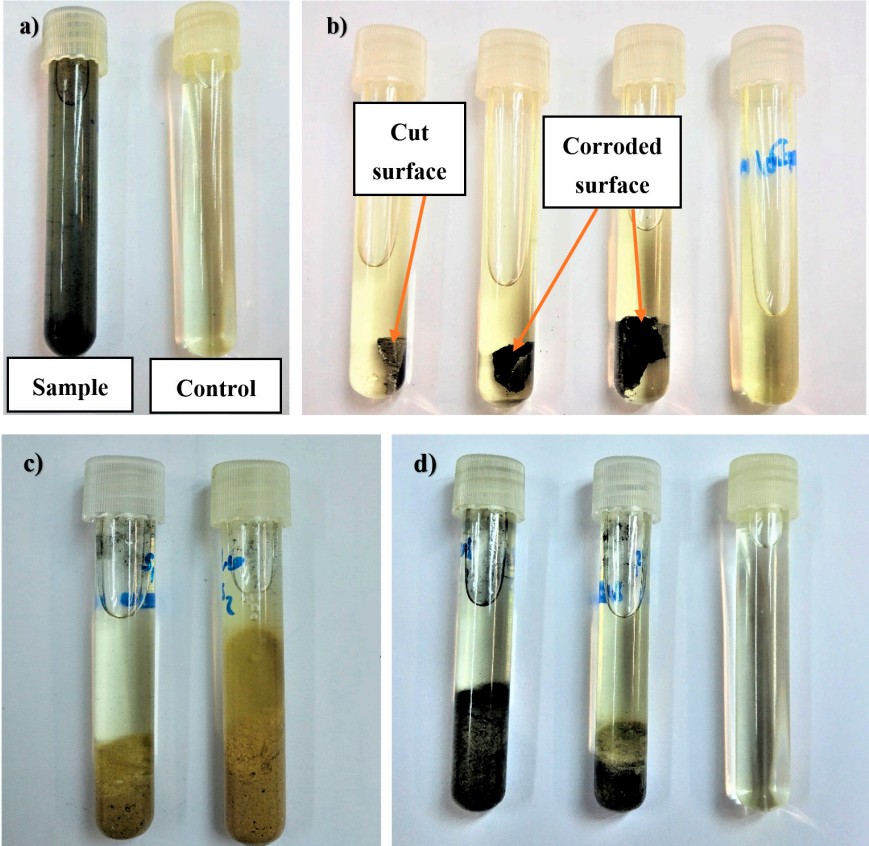

**Figure 11.** SRB growth monitoring in specific media after 20 days. Control: The specific medium without inoculation. (**a**) Surface scratch from corrosion tube, (**b**) the corrosion tube cuts (**c**) the soil samples of corrosion area after five days, and (**d**) after 20 days.

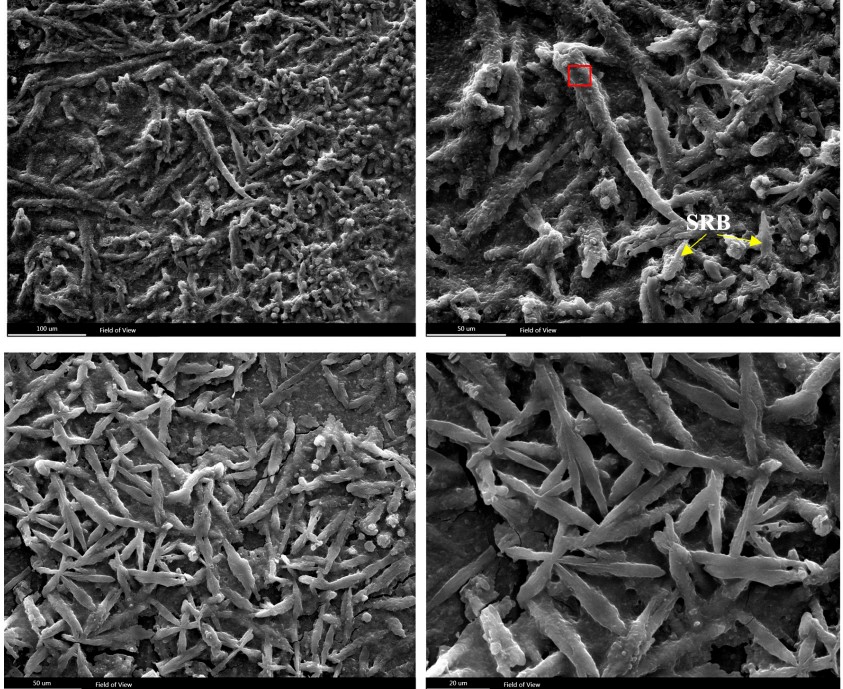

**Figure 12.** SEM images of SRB growth in specific media inoculated by the surface scratches of the corroded sample (all pictures are from similar features with different magnifications).

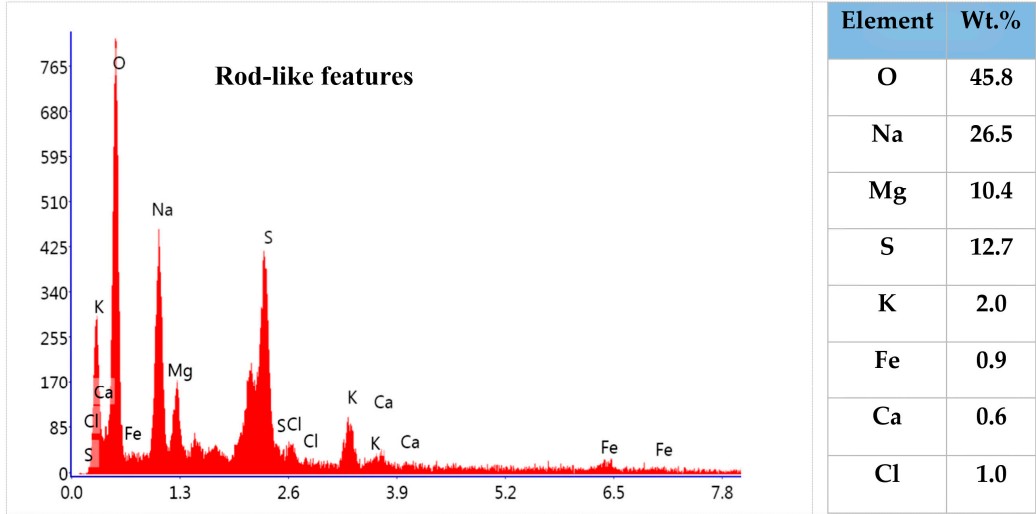

| Element | Wt.% |
|---------|------|
| O | 45.8 |
| Na | 26.5 |
| Mg | 10.4 |
| S | 12.7 |
| K | 2.0 |
| Fe | 0.9 |
| Ca | 0.6 |
| Cl | 1.0 |

**Figure 13.** An EDS analysis of area shown in Figure 12 (*x*-axis: Energy (KeV), *y*-axis: a.u.).

The corroded tube was also cultivated in the media. The results revealed that due to reactivation and reproduction of SRB, the amount of corrosion products increased. The corrosive effects of SRB are shown in Figure 14. The redox potentials of as-received soil before inoculation in an SRB-specific medium and that after inoculation were 19 and −315 mV, respectively. The decrease in the redox potential can be attributed to the enhanced activity of SRB, which is in accordance with the presented results of Figures 11–13 [23,24].

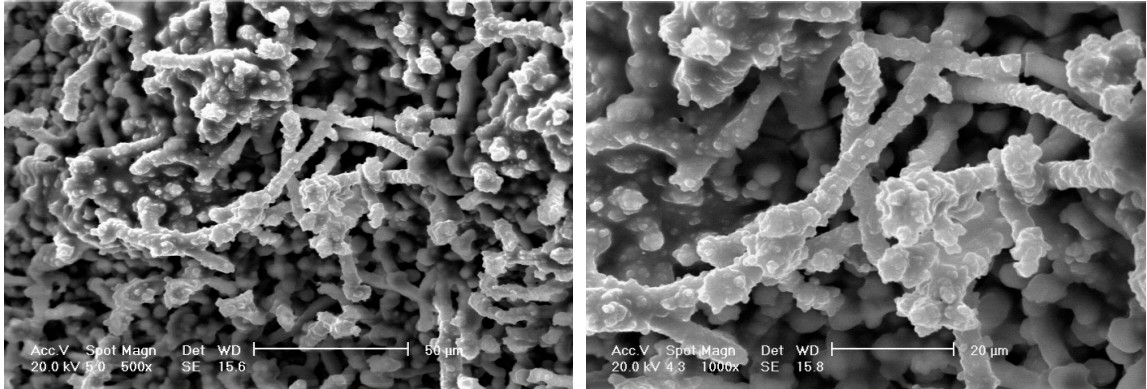

**Figure 14.** Corrosion products development after reactivation of SRB due to inoculation of the corroded tube cuts in the SRB-specific media (Images are from similar features with different magnifications).

## 4. Conclusions

This paper investigates an MIC-induced severe failure of carbon steel elbows in an underground amine pipeline. The following conclusions can be drawn:

(i) The observed failure was in the form of localized pitting corrosion, which originated from the outer surface of the elbow. Observed pits had a different size and morphology, with some being very small, while a few were deep and had already turned into a hole penetrating the pipe wall.

(ii) The corrosion at the inner surface was predominantly in the form of a homogeneous general corrosion. It appears that the corrosion products at the outer surface have three distinctive morphologies: Rod-like morphologies, colonies of spheres, and flower-like morphologies. These features in some cases were as large as a few hundred micrometers.

(iii) The EDS results showed a high concentration of sulfur in the chemistry of these products, inferring that these features are SRB-related products. This was confirmed by SRB inoculation experiments.

(iv)   In order to prove the presence of SRB, corroded parts of elbows and samples of soil close to the damaged areas were inoculated in SRB-specific media and incubated in anaerobic conditions. The black precipitation was created due to SRB growth, confirming the existence of SRB and that it had a corrosion-inducing role in this failure. This was confirmed by microscopy results.

**Author Contributions:** Conceptualization, M.K.K. and A.B.; methodology, M.K.K., A.B. and A.H.-A; formal analysis, M.K.K., A.B. and A.H.-A.; resources, M.K.K., A.H.-A and A.B.; data curation, M.K.K., A.B., A.H.-A. and M.K.; writing—original draft preparation, M.K.K.; writing—review and editing, A.B., M.K.K., A.H.-A., P.T. and M.K.; supervision, A.B. and P.T.; project administration, A.B.

**Funding:** This research received no external funding.

**Conflicts of Interest:** The authors declare no conflicts of interest.

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
