# Peer review of "Microbiologically Influenced Corrosion of a Pipeline in a Petrochemical Plant"

_metals, doi:10.3390/met9040459_

Reviewer 1 Report

The manuscript “Microbiologically Influenced Corrosion of a Pipeline in a Petrochemical Plant” presents a failure analysis of buried carbon steel piping used in petrochemical plant.

General comments

This kind of failure reports from real operating environments have possibility to bring new understanding of complex natural phenomena related to MIC and combining the results to operational aspects of industry. In current form, the manuscript however is not contributing to increasing of scientific understanding of MIC. The authors need to revise the manuscript carefully to ensure that the results are interpreted in light of current scientific understanding. With the careful revision this manuscript has a potential to be of interest for wide audience.

In addition, the experimental section is almost non-existing, please amend the experimental section. How samples were stored after removing from underground? Were they dry? Exposed to oxygen? Were the samples fixed or coated for SEM examination?  OES and EDS should be also in experimental section. Experimental section should also have information of cultivation experiment, what samples and how much were used for inoculant. Figure 11 mentions also soil sample.

Were the samples red/yellow in appearance already immediately after removing underground ore were the samples oxidized during processing?

Specific comments:

L 23 important microorganism agent -> key group of microorganisms

L37 by digging the ground -> by digging from the ground

L47 + archaea

L51-52 All MIC processes eventually influence the anodic/cathodic reactions

L56 Planktonic -> planktonic

L59-60 “It is also well-known that microorganism can use a large amount of chemicals as nutrient sources and oxidize them” microorganisms also reduce many chemicals. “large amount” is not relevant term

L61 bio-corrosion, stick to one term, either MIC or bio-corrosion

L63 “(SRB) are microorganisms which have more detrimental effects.” -> (SRB) are microorganisms which are recognised to have more detrimental effects.

L66-67 “As well, SRB can appear under deposits of soil, water, hydrocarbons, chemicals, etc” revise the sentence

L69 check the references

L69 Fe -> Fe0

L70 presence -> availability

L70-71 “,which take place when the soil acts as a barrier between the steel surface and the external environment” revise sentence

L72 “SRB can use iron element” -> SRB can use elemental iron

L89 placed under the soil -> underground

L95-96 Not only sulfur content but sulfate / sulphide content, also the conductivity should be stated

L92 ion -> ions

L111 peptone 1.2 g, meat extract, should it be peptone meat extract 1.2g?

L147 “Rod-shaped products” revise

L149 "statistically more significant", provide the statistics

L157-158 EDS is not very good in distinguishing the composition of this kind of surface products and I would not conclude that the only difference was the morphology.

L164 revise sentence

L164-165 Cathodic depolarization theory is not scientifically supported anymore, please see more recent studies

L166-167 Read this sentence carefully and correct the things that are not true. SRBs do not use O2 or Fe3+, Sulfate is not reduced to magnetite etc.

L168 not all SRB tolerate oxygen well but some do

L170 FeCO3 use subscript for 3

L171-173 clarify sentence

L183 not all SRBs produce a lot EPS, in fact often there are other microorganisms present that are mainly responsible for EPS production

L185-186 revise. EPS are composed of charged polymers and can significantly contribute to the accumulation of different chemical species inside EPS.

L190 iron oxidehydroxide

L211 & L212 black dye -> black precipitation

L241 Sulfur -> sulfur

Figure 1. schematic -> Schematic

Figure 1. Check the correctness of your statements in schematic view. Cathodic reaction needs to be revised and distinct the reactions taking place on carbon steel and in the biofilm or surrounding environment

Figure 5. I would rather use term “surface deposit” rather than biofilm since obviously many surface formations seen in SEM are abiotic.  

Figure 12. I don’t think that the long 100um rods that arrows are pointing are SRB, instead there are cells resembling SRB morphology in right lower corner

Table 1. add reference for specification limit

Author Response

Reviewer 1:

Thanks very much for your inputs. We tried to address all mentioned issues. With your concerns
being carefully addressed, we believe that the manuscript has been significantly improved.
Thanks a lot for your in-depth and detailed revision. Details of actions we have taken on each of
your comments are explained in here:

General comments

This kind of failure reports from real operating environments have possibility to bring new understanding of complex natural phenomena related to MIC and combining the results to operational aspects of industry. In current form, the manuscript however is not contributing to increasing of scientific understanding of MIC. The authors need to revise the manuscript carefully to ensure that the results are interpreted in light of current scientific understanding. With the careful revision this manuscript has a potential to be of interest for wide audience.

In addition, the experimental section is almost non-existing, please amend the experimental section. How samples were stored after removing from underground? Were they dry? Exposed to oxygen? Were the samples fixed or coated for SEM examination?  OES and EDS should be also in experimental section. Experimental section should also have information of cultivation experiment, what samples and how much were used for inoculant. Figure 11 mentions also soil sample.

Manuscript is modified. We tried to make experimental section more clear and informative. For EDS analysis, samples were not coated by gold/carbon to make sure that results are not affected by the coating. Soil samples were collected from different areas around the pipe.

Were the samples red/yellow in appearance already immediately after removing underground ore were the samples oxidized during processing?

The samples had the same color during and before the process. 

Specific comments:

L 23 important microorganism agent -> key group of microorganisms

It is done. Thanks.

L37 by digging the ground -> by digging from the ground

It is done.

L47 + archaea

Done. Thanks.

L51-52 All MIC processes eventually influence the anodic/cathodic reactions

L56 Planktonic -> planktonic

It’s been taken care of.

L59-60 “It is also well-known that microorganism can use a large amount of chemicals as nutrient sources and oxidize them” microorganisms also reduce many chemicals. “large amount” is not relevant term

Done. Thanks for your attention.

L61 bio-corrosion, stick to one term, either MIC or bio-corrosion

Done.

L63 “(SRB) are microorganisms which have more detrimental effects.” -> (SRB) are microorganisms which are recognized to have more detrimental effects.

It’s been taken care of. Thanks.

L66-67 “As well, SRB can appear under deposits of soil, water, hydrocarbons, chemicals, etc” revise the sentence

Done.

L69 check the references

Done.

L69 Fe -> Fe0

It is done. Many thanks.

L70 presence -> availability

Done.

L70-71 “,which take place when the soil acts as a barrier between the steel surface and the external environment” revise sentence

Done.

L72 “SRB can use iron element” -> SRB can use elemental iron

Done. Thanks.

L89 placed under the soil -> underground

Done.

L95-96 Not only sulfur content but sulfate / sulphide content, also the conductivity should be stated

Done.

L92 ion -> ions

It is done.

L111 peptone 1.2 g, meat extract, should it be peptone meat extract 1.2g?

Done. Thanks.

L147 “Rod-shaped products” revise

It is done.

L149 "statistically more significant", provide the statistics

Due to large variations in size and geometry of features, it is extremely difficult to come up with exact statistical numbers.

L157-158 EDS is not very good in distinguishing the composition of this kind of surface products and I would not conclude that the only difference was the morphology.

We totally agree that EDS is not a very precise method. In this case, it is more used as a semi-quantitative method to compare chemical compositions.

L164 revise sentence

Done.

L164-165 Cathodic depolarization theory is not scientifically supported anymore, please see more recent studies.

Manuscript is modified. Thanks.

L166-167 Read this sentence carefully and correct the things that are not true. SRBs do not use O2 or Fe3+, Sulfate is not reduced to magnetite etc.

We fully agree with you on this issue. The manuscript is modified. I am grateful for the positive

learning comment of yours on this issue.

L168 not all SRB tolerate oxygen well but some do

That’s very generous of you to say. Thanks.

L170 FeCO3 use subscript for 3

Done. Many thanks.

L171-173 clarify sentence

Text is modified. Thanks.

L183 not all SRBs produce a lot EPS, in fact often there are other microorganisms present that are mainly responsible for EPS production

I really appreciate it. Thanks.

L185-186 revise. EPS are composed of charged polymers and can significantly contribute to the accumulation of different chemical species inside EPS.

It is done. Many thanks.

L190 iron oxidehydroxide

It is done.

L211 & L212 black dye -> black precipitation

Done. Many thanks.

L241 Sulfur -> sulfur

Done. Thanks.

Figure 1. schematic -> Schematic

Text is modified. Thanks.

Figure 1. Check the correctness of your statements in schematic view. Cathodic reaction needs to be revised and distinct the reactions taking place on carbon steel and in the biofilm or surrounding environment

Text is modified. I appreciate your comment.

Figure 5. I would rather use term “surface deposit” rather than biofilm since obviously many surface formations seen in SEM are abiotic. 

We fully agree with you on this issue. The manuscript is modified.

Figure 12. I don’t think that the long 100um rods that arrows are pointing are SRB, instead there are cells resembling SRB morphology in right lower corner

Figure is modified.

Table 1. add reference for specification limit

Done. Thanks.

Reviewer 2 Report

The manuscript examines a practical corrosion problem related to microbial corrosion. Study is well conducted and obtained results are in agreement with provided conclusions. There are only some typographical and language errors to be corrected. Also please provide references for statements in lines 166-167.

Author Response

Reviewer 2:

Thanks very much for your inputs. We tried to address all mentioned issues. With your concerns
being carefully addressed, we believe that the manuscript has been significantly improved.

The manuscript examines a practical corrosion problem related to microbial corrosion. Study is well conducted and obtained results are in agreement with provided conclusions. There are only some typographical and language errors to be corrected. Also please provide references for statements in lines 166-167.

Manuscript is modified based on your comment. Many thanks.

Reviewer 3 Report

The paper needs significant completion especially with microbiological results. It is necessary to mention the cell number of the anaerobes and the aerobes, too, as in case of the metal sample got form the industrial pipeline, which was covered by soil, should contain all types of microorganisms.

It should make wider the Introduction as it should contain more information about the aerobic and anaerobic metal corrosion.

The sulfate reducing bacteria cover a wide range of microorganisms. One of the most important groups is the Desulfovibrios.

The Authors do mention that the hydrogen sulfide formed by sulfate reducing microorganisms play a role in the biogenic sulfide corrosion in crude oil, but do not mention that some sulfate reducers can oxidize in anaerobic reaction the methane:

CH4 + SO42- ® HCO3- + HS- + H2O

On the SEM images it is not clear why do the Authors state that the long worm-like formations cover the sulfate reducers. The shape and the length are not typical for these microorganisms.

It is interesting that the MIC type corrosion was observed only on the outer part of the tube under investigation but not inside. When big holes were formed (like in the case the Authors discuss) that allowed the penetration of the aqueous liquid contaminated with microorganisms (among them with SRB) how is it possible?

Line 15: amine pipeline; Line 77: amine plant installation. “Please, give some information about this type of pipeline.

Line 55: Instead of “loads of attentions” most probable “lots of attentions”

Line 56: “Planktonic cells” The planktonic is not written with capital letter.

Line 57:  “at last produce an extracellular polymer substance (EPS).” The EPS production starts much earlier, this slimy layer surrounds the microbes already in the planktonic form of the microorganisms!

Line 60: “Interestingly???, anaerobic microorganisms can also grow and control corrosion, in the absence of oxygen.” Per definition the aerobes do not need oxygen for their life.

Line 73: “Cathode” starts with small letter.

Line 74: “nebtioned reactions”???

Line 77: “loads of studies” lots of studies

In Table 1: SO3? Not sulfate ion SO42-? Please, inform the reader about the plasticity index.

Line 108: Nital 60%.” As not only metallurgists will read this paper, please, give information about the Nital (solution of nitric acid with alcohol for eching…)

Line 113: “Liquid” starts with small letter.

Line 117: “Both inner and outer surfaces are covered 117 with red/yellow-brown rust layer.” There was not any sign of blackish deposition caused by the presence of FeS? This would have shown the activity of anaerobic sulfate reducers.

Line 148: “iii) flowerlike surface features, all shown in Figure 6.” This is Figure 5 d.

Line 167: “(4) reduce sulfate to magnetite.” How is possible to reduce sulfate to an oxide mineral Fe(II)Fe(III)O4?

Line 171: “what is important other than the nature of microorganisms,…” Please, rephrase the whole sentence.

Author Response

Reviewer 3:

Thanks very much for your inputs. We tried to address all mentioned issues. With your concerns being carefully addressed, we believe that the manuscript has been significantly improved. Thanks a lot for your in-depth and detailed revision. Details of actions we have taken on each of your comments are explained in here:

The paper needs significant completion especially with microbiological results. It is necessary to mention the cell number of the anaerobes and the aerobes, too, as in case of the metal sample got form the industrial pipeline, which was covered by soil, should contain all types of microorganisms.

Many thanks for your comment. Concerning the exact cell numbers, this is more an industrial failure case study rather than a laboratory investigation. Experimental data in this case is used to find out the root cause of failure. Further in-depth investigation is being done. Results will be published in another paper.

It should make wider the Introduction as it should contain more information about the aerobic and anaerobic metal corrosion. The sulfate reducing bacteria cover a wide range of microorganisms. One of the most important groups is the Desulfovibrios.

Manuscript is modified based on your comment. Thanks.

The Authors do mention that the hydrogen sulfide formed by sulfate reducing microorganisms play a role in the biogenic sulfide corrosion in crude oil, but do not mention that some sulfate reducers can oxidize in anaerobic reaction the methane:

CH4 + SO42- ® HCO3- + HS- + H2O

This added to the manuscript. Thanks

On the SEM images it is not clear why do the Authors state that the long worm-like formations cover the sulfate reducers. The shape and the length are not typical for these microorganisms.

Some new SEM images are added to the paper, with features shown below. EDS analyses confirm that these features are SRB.

Fig. 1. SEM images of SRB

It is interesting that the MIC type corrosion was observed only on the outer part of the tube under investigation but not inside. When big holes were formed (like in the case the Authors discuss) that allowed the penetration of the aqueous liquid contaminated with microorganisms (among them with SRB) how is it possible?

We believe that MIC has been an active mechanism at the interface of soil and outer surface of the pipe. In the course of time MIC has led to localized corrosion and pitting on the outer surface. So, there is no penetration of liquid. 

Line 15: amine pipeline; Line 77: amine plant installation. “Please, give some information about this type of pipeline.

Some information is added to the manuscript. Thanks for your comment.

Line 55: Instead of “loads of attentions” most probable “lots of attentions”

Done.

Line 56: “Planktonic cells” The planktonic is not written with capital letter.

Done. Many thanks.

Line 57:  “at last produce an extracellular polymer substance (EPS).” The EPS production starts much earlier, this slimy layer surrounds the microbes already in the planktonic form of the microorganisms!

Done.

Line 60: “Interestingly???, anaerobic microorganisms can also grow and control corrosion, in the absence of oxygen.” Per definition the aerobes do not need oxygen for their life.

Text is modified. Thanks.

Line 73: “Cathode” starts with small letter.

It is done.

Line 74: “nebtioned reactions”???

Text is modified.

Line 77: “loads of studies” lots of studies

It is done. Many thanks.

In Table 1: SO3? Not sulfate ion SO42-? Please, inform the reader about the plasticity index.

It is a report from laboratory of the plant according to ASTM D4318 - 17e1 and ASTM D4972 – 18. Please see fig. B. The plasticity index (PI) is a measure of the plasticity of a soil. The plasticity index is the size of the range of water contents where the soil exhibits plastic properties.

Fig. 2. Report of specifications of the onsite soil.

Line 108: Nital 60%.” As not only metallurgists will read this paper, please, give information about the Nital (solution of nitric acid with alcohol for eching…)

It is done. Many thanks.

Line 113: “Liquid” starts with small letter.

Done.

Line 117: “Both inner and outer surfaces are covered 117 with red/yellow-brown rust layer.” There was not any sign of blackish deposition caused by the presence of FeS? This would have shown the activity of anaerobic sulfate reducers.

Indeed, we observed blackish deposits all over the outer surface as well. Please see the following image:

Line 148: “iii) flowerlike surface features, all shown in Figure 6.” This is Figure 5 d.

It is done. I appreciate your comment.

Line 167: “(4) reduce sulfate to magnetite.” How is possible to reduce sulfate to an oxide mineral Fe(II)Fe(III)O4?

Text is modified. Thanks.

Line 171: “what is important other than the nature of microorganisms,…” Please, rephrase the whole sentence.

Text is modified. Many thanks.

Reviewer 4 Report

The manuscript deals with microbiologically induced corrosion in metal pipelines of petrochemical plants. Therefore, it targets a subject of current relevance that falls within the scope of this journal.

In this reviewer's opinion, the text is sufficiently clear and, hence, the work can be accepted after a MINOR REVISION intended at addressing the comments highlighted throughout the marked manuscript attached at the present review report.

Author Response

Reviewer 3:

Thanks very much for your inputs. We tried to address all mentioned issues. With your concerns
being carefully addressed, we believe that the manuscript has been significantly improved.

I would suggest merging sections 2 and 3 in a unique section (customarily) entitled "Materials and methods".

It is done. Thanks.

itemise the three points with the aim to enhance readability.

It is done. I appreciate your comment.

This section should be completely rewritten. After an introductory sentence, the main findings have to be listed in concise items.

Please, do not refer to mere empirical evidence, but try to point out the mechanical reasons behind them.

Manuscript is modified. Many thanks.

Round  2

Reviewer 1 Report

For most part the manuscript has improved with the revisions made by the authors.

However, some instances should still be corrected.

Figure 1. 

At the anode the Fe2+ should be written so that it is released from the steel

electron flow from the anode to cathode happens in steel matrix

At the cathode the electrons meet 2H+ and H2 is released (as SRBs are thought to function under anoxic conditions)

reduction of SO42- happens inside the biofilm or in close vicinity to the pipe not in steel matrix.

Table 1. the reference added "taken from the laboratory of the plant" is not sufficient, is this done according to the some standards? Maybe you can refer to those?

Experimental section:

Add OES analysis details

Soil sampling, how many soil samples, distance to failure site, sampling size, conditions of storage before cultivation

L253 hole penetraiting the pipe wall.

Conclusions section could also highlight that in the light of results (microscopy, surface analysis and cultivation) the authors suspect that the SRBs originating from surrounding soil have had a corrosion inducing role.

Author Response

Reviewer 1:

Dear Reviewer,

Thanks a lot for your in-depth and detailed revision. Details of actions we have taken on each of your comments are explained in here:

Figure 1. At the anode the Fe2+ should be written so that it is released from the steel electron flow from the anode to cathode happens in steel matrix. At the cathode the electrons meet 2H+ and H2 is released (as SRBs are thought to function under anoxic conditions) reduction of SO42- happens inside the biofilm or in close vicinity to the pipe not in steel matrix.

Figure 1 is revised, based on your explanation. 

Table 1. the reference added "taken from the laboratory of the plant" is not sufficient, is this done according to the some standards? Maybe you can refer to those?

References to standards are added to the table caption.

Experimental section:

Add OES analysis details

It is done. Many thanks.

Soil sampling, how many soil samples, distance to failure site, sampling size, conditions of storage before cultivation

Soil samples were collected from the vicinity of damaged area. Samples were collected from ten different spots with approximate weight of 10 g each. Collected soil samples were stored in plastic bottles.

L253 hole penetraiting the pipe wall.

Done. Thanks.

Conclusions section could also highlight that in the light of results (microscopy, surface analysis and cultivation) the authors suspect that the SRBs originating from surrounding soil have had a corrosion inducing role.

It is done.
